# Microgrids Power Quality Enhancement Using Model Predictive Control

Felix Garcia-Torres [1,*], Sergio Vazquez [2], Isabel M. Moreno-Garcia [3], Aurora Gil-de-Castro [3], Pedro Roncero-Sanchez [4] and Antonio Moreno-Munoz [3]

1   Application Unit, Centro Nacional del Hidrogeno, 13500 Puertollano, Spain
2   Electronic Department, Universidad de Sevilla, 41092 Sevilla, Spain; sergi@us.es
3   Electronic and Computer Department, Escuela Politécnica Superior, Campus de Rabanales, Universidadde Córdoba, Edificio Leonardo da Vinci, 14071 Córdoba, Spain; isabel.moreno@uco.es (I.M.M.-G.); agil@uco.es (A.G.-d.-C.); amoreno@uco.es (A.M.-M.)
4   Systems Engineering and Automatic Control Department, University of Castilla-La Mancha, 13001 Ciudad Real, Spain; Pedro.Roncero@uclm.es
*   Correspondence: felix.garcia@cnh2.es; Tel.: +34-926-42-06-82

**Abstract:** In electric power systems, any deviation with respect to the theoretical sinusoidal waveform is considered to be a disturbance in the power quality of the electrical grid. The deviation can alter any of the parameters of the waveform: frequency, amplitude, and symmetry among phases. Microgrid, as a part of the electric power system, has to contribute providing an adequate current waveform in grid connected-mode, as well as to guarantee similar voltage features than the standard requirement given for public distribution grids under normal exploitation conditions in islanded mode. Adequate power quality supply is necessary for the correct compatibility between all the devices connected to the same grid. In this paper, the power quality of microgrids is managed using a Model Predictive Control (MPC) methodology which regulates the power converters of the microgrids in order to achieve the requirements. The control algorithm is developed for the following microgrids working modes: grid-connected, islanded, and interconnected. The simulation results demonstrate that the proposed methodology improves the transient response in comparison with classical methods in all the working modes, minimizing the harmonic content in the current and the voltage even with the presence of non-balanced and non-harmonic-free three-phase voltage and current systems.

**Keywords:** microgrids; power quality and reliability; Model Predictive Control; interconnected systems; harmonics; power system control



## 1. Introduction

Power quality and reliability (PQR) will be important factors in the transition towards the smart grid. In accordance with the different national policies, the generation should meet the growing demands of cleanly, reliability, sustainability, and low cost [1]. Depending on the power quality perturbation grade and the sensitivity of the receptors, it may have a repercussion on other devices. Adequate quality supply provides the necessary compatibility between all the devices connected to the same grid. In traditional power systems, power quality in a node of the grid is associated with the short circuit power at this point of the grid. Under constant emission, higher short circuit power results in a better voltage quality. The controllability of fossil-fuel power plants on which the centralized generation has been based, as well as the traditional lineal loads just cause low level power quality anomalies in comparison with short circuit power in the upstream network [1]. But this scenario has been modified in recent years marking the transition to the smart grid with the introduction of distributed generators and electronic loads. Regardless of the scenario, power quality would be defined by the interaction between the generation equipment,

the consumer devices and the grid. All anomalies should be compensated by the Energy Storage Systems (ESS). Microgrids can be seen as a key technology to improve PQR aspects of future smart grids. Their ability to work in grid-connected or islanded mode is specially adequate to supply electricity to sensitive loads. In this scenario, enhanced power quality operation of microgrids should be included developing advanced power electronics for interfacing ESS, which minimizes the intermittent effects of renewable energy systems and compensates for the presence of harmonics or unbalanced loads. The controller of these microgrids should include fast transition between grid-connected and islanded mode in order to mitigate the effects of faults in the main grid. Microgrids will also be characterized by a high share of power-electronics devices, increasing thus harmonic levels and possibly causing instabilities owing to interactions between controllers and resonances.

### 1.1. Literature Review

The microgrids control systems must address several aspects with different timescales from their optimization in the day-ahead electrical markets, with sample periods of 1 h, towards real-time control of PQR issues, with $T_s < 1$ s, for which requirements involve different control approaches and different time scales. Fast electrical control of the phase, frequency and voltage of individual resources must be carried out in time scales lower than second or less, while unit commitment, economic dispatch, demand-side optimization and energy exchanges with the utility grid are performed with longer time scales (minutes or hours). Thus, an extended approach is to develop a hierarchical control structure [2]. The major issues and challenges in the microgrid control are discussed in Reference [3], highlighting the given challenges for the control of PQR in microgrids. This topic is deeply studied in the review paper carried out in Reference [4].

PQR aspects of microgrids can be divided into the primary and secondary control levels of microgrids. In the primary control level, the voltage and frequency delivered by the inner loop of each inverter are regulated. Droop control is most commonly used at primary control level: This method assigns to each inverter of the microgrid a droop characteristic based on its generation capabilities [5]. Conventional droop control considers that line impedance is inductive but this assumption is not correct because the output impedance is dependent on the control strategy applied when using power electronics devices. Advanced droop methods are presented in several papers [6,7]. These methods have the drawback of active and reactive power coupling, which has been addressed by several authors making an approach based on the virtual output impedance method. As a result, the expected voltage can be modified [8]. Secondary control system is usually required to correct the frequency and voltage. Additionally, secondary control algorithms can be used for reactive power compensation [9] and to reduce the harmonics content of the voltage waveform [10]. Most of the existing literature for primary and secondary control in microgrids is based on classical PI-PWM controllers. These kinds of controllers do not achieve good results in the transient response, which is highly dependent of the tuning of the corresponding parameters of the controllers.

Model predictive control (MPC) presents several attractive features to be applied in the PQR management of microgrids, appearing as a powerful tool to overcome some of the previously commented problems. The controller can take into account the future behavior of the power inverter despite its complex dynamics. The cost function can integrate multiple criteria, allowing the optimization of important parameters, such a active and reactive power control, harmonics reduction, or ripple minimization. MPC can easily manage the transition between islanded and grid-connected modes, achieving a faster response than the one obtained by classical PI-PWM controllers. This aspect can be crucial in case of energy supply to critical loads. The field of MPC in power converters has been applied from two main control strategies: Finite Control Set (FCS) and Continuous Control Set (CCS).

The FCS-MPC methodology is based on the finite number of switching states that a power inverter can adopt. The optimization problem is simplified with the prediction of

the converter behavior considering these possible switching states. Every time that the controller runs, the set of admissible switching sequences are numbered, thus predicting the corresponding system response based on a prediction model and evaluating the cost function according to the prediction carried out. The controller applies to the system the control sequence, which yields the minimal value in the cost function. Therefore, the cost function is minimized using the Exhaustive Searching Algorithm method [11]. The CCS method generates continuous-time signals as control actions, which are sent to a modulator, and the optimization problem is solved analytically by setting the derivative of the cost function equal to zero in the unconstrained case. Its main advantage is the use of longer control horizons, since an analytical solution is provided. Nevertheless, with complex topologies of power inverters this methodology presents difficulties to create an appropriate model of the plant, being also necessary higher computational resources. A basic application for this kind of controllers can be found in Reference [12].

The last studies related to power quality enhancement with power inverters search the harmonic compensation using four-leg voltage source inverter topologies with active neutral control. In Reference [13], an MPC control strategy based on the optimal switching sequence concept for a single-phase grid-connected H-bridge neutral-point-clamped (H-NPC) power converter is presented. In Reference [14] an active power filter implemented with a four-leg voltage-source inverter (VSI) using an MPC scheme is presented for grid-connected applications. The paper presented in Reference [15] applies similar methodology in the new implementation of the finite control states set model predictive control (FS-MPC) applied to three-level four-leg flying capacitor converter (FCC), operating as a shunt active power filter. The obtained results by the control algorithm improves the current tracking capability and the transient response. Nevertheless, the use of FCS-MPC controller has poor results in the THD content.

In Reference [16], a cascade-free fuzzy FCS-MPC is proposed for neutral point-clamped power inverters with low switching frequency (SF). The main objective of the proposed method is to achieve a low SF operation. The cost function is formulated to reduce the SF, and a fuzzy logic control (FLC) technique is employed to dynamically choose the weighting factors. The article presented in Reference [17] proposes a novel flexible reference current generation technique by using a tuning parameter to reduce the active power oscillation flexibly. The generated reference current comprises not only the positive and negative sequence currents, but also lower order harmonic components. A flexible multi-frequency reference current computation technique for the unbalanced and distorted grid conditions is developed using MPC control techniques. The experimental and simulation results successfully validated the trade off between the low frequency power oscillations and the current THD, which was established using a tuning parameter. Therefore, with the proposed scheme active power oscillations can be reduced in microgrids scenarios.

In Reference [18], a composite selective harmonic elimination pulse-width modulation (SHE-PWM) and MPC for seven-level hybrid-clamped (7L-HC) inverters is presented. The proposed methodology achieves as results low a SF with good harmonic performance with a reduced computational burden. In Reference [19], a strategy that combines FCS-MPC with SHE modulation pattern in its formulation is proposed to govern multilevel power converters. The proposed methodology is based on considering a desired operating point for the system state (converter current reference), an associated predefined SHE voltage pattern is obtained as a required steady-state control input reference. Then, the cost function is formulated with the inclusion of both system state and control input references. The obtained experimental results present a fast dynamic response, while a predefined voltage and current spectrum with low SF is achieved in steady-state. In Reference [20], a model predictive power control (MPPC) scheme and a model predictive voltage control (MPVC) scheme are presented. The proposed methodology consists in controlling the bidirectional buck-boost converters of the battery ESS based on the MPPC algorithm, the fluctuating output from the renewable energy sources can be smoothed, while stable dc-bus voltages can be maintained as the inverters inputs. Then, the parallel inverters

are controlled by using a combination of the MPVC scheme and the droop method to ensure a stable AC voltage output and a proper power sharing. Compared with the traditional cascade control, the proposed method is simpler and shows better performance, which is validated when simulated with MATLAB/Simulink and on Real-Time Laboratory (RT-LAB) platform.

Microgrids consist of multiple parallel-connected distributed generators, storage devices, or controllable loads which are able to operate in both grid-connected and islanded modes, in a coordinate mode. In islanded mode, it is required to maintain system stability and power quality among the multiple parallel interconnected devices. Deficit balances in the active and reactive power between the different components of the microgrids, due to several aspects, such as the influence of impedance mismatch of the feeders and the different ratings of the distributed units, can lead to poor power quality indexes, which can damage the connected devices to the same microgrid AC/DC bus. In grid-connected mode, imbalances in the active/reactive power can affect to the schedule carried out with the main grid in the tertiary control. The increasing presence of non-linear loads and unbalanced loads could further affect to the global power balance in the microgrid. The importance of the management of power losses and power quality degradation due to the circulating current in interconnected microgrids (e.g., hybrid AC/DC microgrids) is studied in Reference [21,22].

In order to coordinate the different generators integrated in the microgrid, different secondary control algorithms have been proposed. In Reference [23], an MPC-based controller and a Smith predictor (SP) based controller are applied to the secondary level of the microgrid. The results of this work prove that, with the proposed methodology, the nominal values of frequency can be reached with a faster speed but fewer oscillations during load variations. In addition, the MPC-controller with the SP solves the problems brought by the communication delays. In Reference [24], a fuzzy adaptive model predictive approach for load frequency control of an isolated microgrid is proposed. The frequency deviation problem is solved using a centralized MPC, which is made adaptable by dynamically adjusting its parameters using a fuzzy controller. In Reference [25], a distributed secondary control scheme for both voltage and frequency control in autonomous microgrids is shown. The algorithm incorporates predictive mechanisms into distributed generations, the secondary voltage control is converted to a tracker consensus problem of a distributed model predictive control, with the synchronous convergence procedure for voltage magnitudes to the reference value drastically accelerated at a low communication cost. In Reference [26], a virtual inertia control-based MPC for microgrid frequency stabilization is developed. In Reference [27], a Distributed Model Predictive Control (DMPC)-based strategy for regulating the frequency and average voltage and achieving real and reactive power consensus in the microgrid is presented.

*1.2. Main Contributions*

In AC microgrids, the final power quality obtained in the microgrid depends on the exchanged power flow between its local devices connected and the grid. An appropriate ESS connected to a VSI can be used to enhance the power quality of the microgrid. The final result not only depends on the topology of the VSI, but also on its control system. With the aim to obtain an optimal power flow in the microgrid, a four wire VSI with active neutral control is selected in order to integrate unbalanced and non-linear loads. The VSI response is improved using an innovative MPC controller to manage the power quality of the microgrids and their power exchange with the main grid or with the neighbor microgrids. The studied references in the previous section do not solve completely the problem of the transient response due to the fact that droop controls and/or PI-based controller are applied. The state of the art of the MPC controllers applied to primary control of the microgrid uses the instantaneous expressions for active and reactive power expressed with Park/Clark's transformation. These expressions can only be computed for balanced and harmonic-free three-phase voltages. Based on the MPC strategy which uses Fourier's

transform presented in Reference [28], this paper develops a primary MPC using the expression for active and reactive powers of the voltage-current pairs calculated at the fundamental frequency. The presented method for the voltage control of the microgrid in Reference [28] is extended here to include non-linear and non-balanced loads, developing an MPC-controller applied to four-wire three-phase voltage source inverters with active control of the neutral point. The method is also expanded to include the islanded and grid-connected operation modes. Different case studies to demonstrate the enhanced power quality operation of the microgrid, such as harmonic mitigation, unbalanced, and non-linear loads in both modes are shown. The behavior of the control algorithm to achieve a fast transition from grid-connected to islanded mode, and vice versa, is presented. The method is also expanded to be used in the case of interconnected microgrids working under a blackout of the main grid.

The paper is organized as follows: Section 2 provides a description of the controller design, while Section 3 presents the results for the different case studies. Finally, Section 4 outlines the conclusions.

## 2. Controller Design

The topology of the interconnected microgrids object of this paper is shown in Figure 1. As can be seen in Figure 1, there are three intelligent power switches (IPS) installed to isolate or connect the working mode of each microgrid with the main grid and/or with the neighbor microgrid. In AC microgrids, the final power quality obtained in the microgrid depends on the exchanged power flow between its local devices connected and the grid. An appropriate ESS connected to a VSI can be used to enhance the power quality of the microgrid. The final result not only depends on the topology of the VSI, but also on its control system. With the aim to obtain an optimal power flow in the microgrid, a four-wire two levels (2L-VSI) with active neutral control is selected in order to integrate unbalanced and non-linear loads. The power inverter is composed of four-legs $(a, b, c, n)$, composed each one of two ideal power switches, one connected to the positive terminal of the dc voltage source and the other switch is connected to the negative terminal. The switches states are a function of the associated gate-signals. The gate signals applied to the switches placed at a same leg of the inverter are related, being the value of the negative terminal connected gate signal the opposite of the positive terminal gate signal. Each gate signal just can adopt two values: "0" if the power switch is at OFF-state and "1" when the power switch is at ON-state. In order to integrate the possibility to manage unbalanced and non-linear loads, the neutral point of the power inverter is connected to two similar capacitors $C_+$ and $C_-$. The voltage of $C_+$ and $C_-$ is controlled with the power switches $S_{1,n}$ connected to the positive terminal of the $V_{dc}$ source and a common point with $S_{2,n}$, which is connected to the negative terminal. The common point between $S_{1,n}$ and $S_{2,n}$ is connected with the neutral point of the 2L-VSI through a non-ideal inductance composed of $R_{L_N}$ and $L_N$.

The power inverter is connected in the Point of Common Coupling (PCC) of the microgrid with the power grid and the rest of components of the microgrid by means of an LC filter, where $C_f$ is the capacitor of the filter and $L_f$ is the inductance of the filter. In order to integrate the non-ideal behavior of these components, a small $R_{L_f}$ is the series resistance of the inductance, while $R_{C_f}$ is the series resistance of the capacitor $C_f$. The power inverter feeds a microgrid composed of an unbalanced load, a non-linear load, and a bidirectional inverter. The unbalanced load is composed of a resistor $R_a$ in the phase $a$, a non-ideal inductance formed by $R_b$ and $L_b$ in phase $b$, and finally it has got connected a non-ideal capacitor in phase c, in which components are $R_c$ and $C_c$. The non-linear load is formed by an uncontrolled rectifier and a load on the DC side formed by a capacitor filter $C_{non}$ and the resistor $R_{non}$. The rectifier is connected to the PCC of the microgrid with the line impedances given by $L_{non}$. Finally, an AC/DC bidirectional 2L-VSI with LC filter is also included, connected to the microgrid through the line impedance given by $R_{inv}$ and $L_{inv}$. The VSI response is improved using an innovative MPC controller to manage the power quality of the microgrids and their power exchange with the main grid or with the

neighbor microgrids. The block diagram of the controller is exposed in Figure 2. Note that the current control hardware platform provides a high computational capability which makes the proposed method feasible as can be observed in Reference [29].

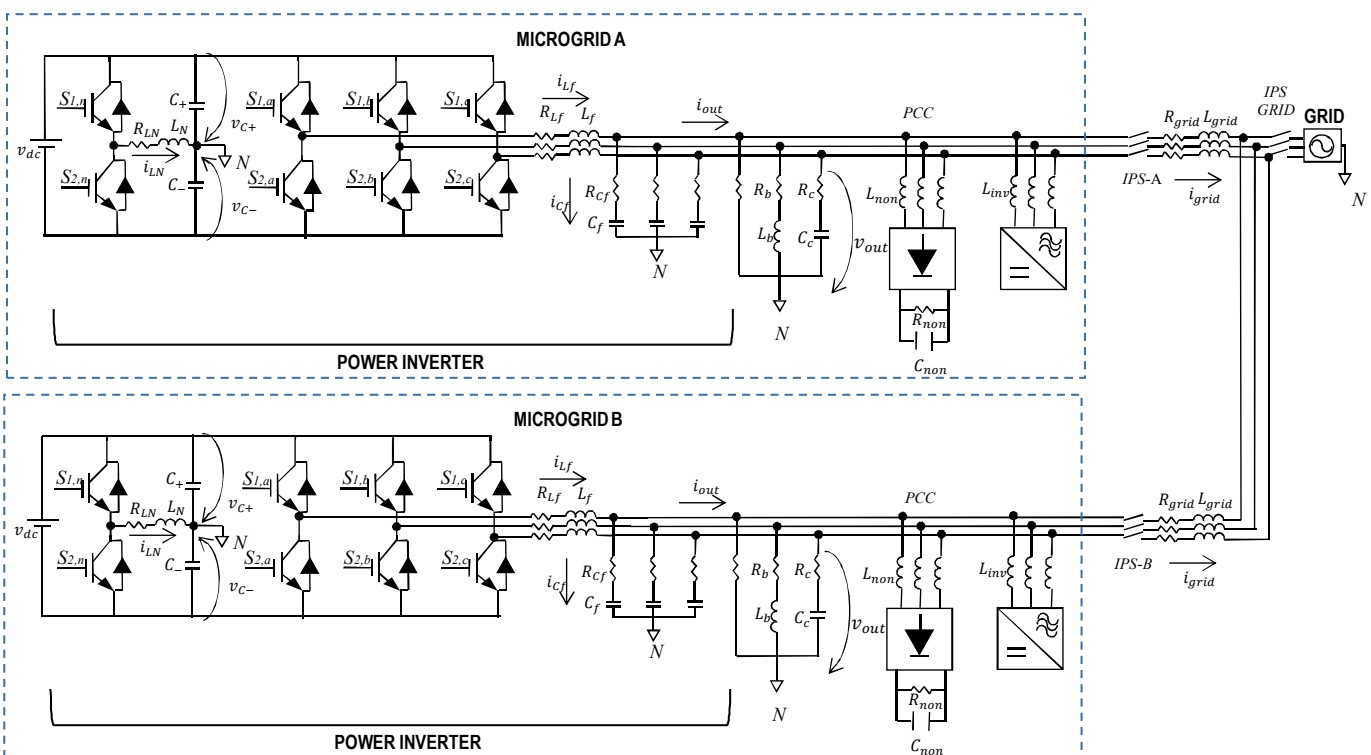

**Figure 1.** Interconnected microgrids object of study.

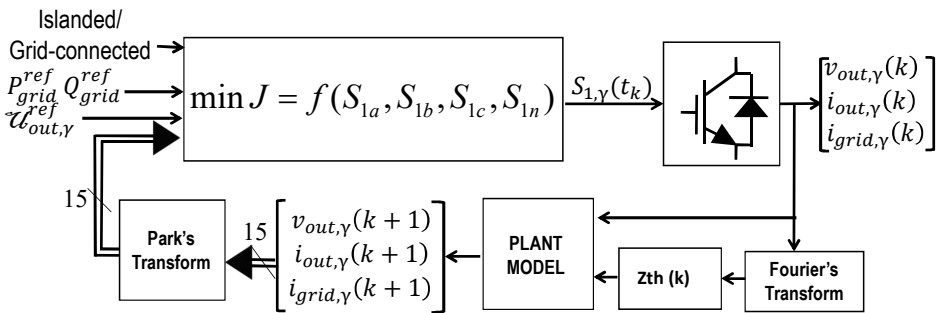

**Figure 2.** Block diagram.

The first step of the controller is to calculate the Fourier analysis of the current and voltage output at the current sample instant *k*. With these measurements the Thevenin's equivalent impedance is calculated at the output of the inverter. With this equivalent impedance the output current and the voltage predictions are carried out, which are included in the cost function to be minimized. Finally, the optimal gate-signal combination is calculated in the FCS-MPC controller.

### 2.1. Fourier Expressions

As explained in the previous section, the instantaneous active power and reactive power expressed with Park's transformation can only be computed for balanced and harmonic-free three-phase voltages, while it is still necessary to compute the active and reactive powers associated with a periodic set of three-phase voltages and currents equations.

This is done through the Fourier analysis of the current and voltage signals per phase. A signal $y(t)$ can be expressed by a Fourier series of the form:

$$y(t) = \frac{a_0}{2} + \sum_{n=0}^{\infty} a_n \cos(n\omega t) + b_n \sin(n\omega t), \tag{1}$$

where $n$ represents the rank of the harmonics ($n = 1$ corresponds to the fundamental component). The magnitude and the phase of the selected harmonic component can be calculated by the next equations:

$$|\mathscr{Y}_n| = \sqrt{a_n^2 + b_n^2}; \quad \angle \mathscr{Y}_n = \arctan\left(\frac{b_n}{a_n}\right), \tag{2}$$

where $\mathscr{Y}_n$ is the Fourier's expression of the signal y(t), which can be expressed in cartesian coordinates with the following expressions:

$$\mathrm{Im}(\mathscr{Y}_n(t)) = a_n^y = \frac{2}{T} \int_{t-T}^{t} y(t) \cos(n\omega t) dt; \tag{3}$$

$$\mathrm{Re}(\mathscr{Y}_n(t)) = b_n^y = \frac{2}{T} \int_{t-T}^{t} y(t) \sin(n\omega t) dt, \tag{4}$$

with $T = 1/f$ being the corresponding period to the fundamental frequency. The upper index $y$ is related to the signal $y(t)$ in which the Fourier analysis is developed. Using the discrete expressions of Equations (2)–(4) with a sample period $T_s$, for the voltage and current signals, the value of this signal expressed in the Fourier's domain $\mathscr{U}(k)$ and $\mathscr{I}(k)$ can be obtained. The equivalent Thevenin's impedance calculated for the fundamental frequency can be estimated in polar coordinates:

$$\left|Z_n^{th}(k)\right| = \frac{\sqrt{(a_n^v(k))^2 + (b_n^v(k))^2}}{\sqrt{(a_n^i(k))^2 + (b_n^i(k))^2}}, \tag{5}$$

$$\varphi_n^{th}(k) = \angle Z_n^{th}(k) = \arctan\left(\frac{b_n^v(k)}{a_n^v(k)}\right) - \arctan\left(\frac{b_n^i(k)}{a_n^i(k)}\right). \tag{6}$$

The upper indexes $v$ and $i$ are related to the output voltage and output current of the inverter, respectively. By expressing $Z_{th,n}(k)$ in cartesian coordinates, the equivalent resistance and the equivalent impedance can be obtained:

$$Z_n^{th}(k) = R_n^{th}(k) + jX_n^{th}(k). \tag{7}$$

Particularizing for $n = 1$, depending of the value of $X^{th}(k)$ an equivalent inductance or capacitance can be obtained, in these cases when $\mathrm{sign}(X^{th}(k)) = \mathrm{sign}(R^{th}(k)))$ expression (8) and using the relationship given in (9) when $\mathrm{sign}(X^{th}(k)) \neq \mathrm{sign}(R^{th}(k))$:

$$L^{th}(k) = \frac{X^{th}(k)}{2\pi f}; \quad C^{th}(k) = 0, \tag{8}$$

$$L^{th}(k) = 0; \quad C^{th}(k) = -X^{th}(k) \cdot 2\pi f. \tag{9}$$

The expression for active and reactive powers of the voltage-current pairs calculated at fundamental frequency can be computed (being $\varphi_{j,n}^{th}(k+1)$ the phase of Thevenin's impedance evaluated at $n = 1$):

$$P(k) = \frac{|\mathscr{U}(k)||\mathscr{I}(k)|}{2} \cos(\varphi_1^{th}(k)), \tag{10}$$

$$Q(k) = \frac{|\mathscr{U}(k)||\mathscr{I}(k)|}{2} \sin(\varphi_1^{th}(k)). \tag{11}$$

Expressions (10) and (11) are valid for all the cases, including non-balanced and non-harmonic-free three-phase voltages and currents systems.

### 2.2. Predictive Model of the VSI

The model of the plant can be obtained as a function of its decision variables (gate signals of each leg $S_{1a}$, $S_{1b}$, $S_{1c}$, and $S_{1n}$), the set of state variables composed by the inductor currents or capacitor voltages and the output currents and voltages of the inverter per phase $[v_{C+}, v_{C-}, i_{LN}, v_{Cf}, i_{Lf}, i_{out,j}, v_{out,j}]$, following the expressions (12)–(17). (Notice that $\Delta y(k+1) = y(k+1) - y(k)$.)

$$v_{dc}(k+1) = +v_{C+,N}(k+1) - v_{C-,N}(k+1), \tag{12}$$

$$i_{Cj}(k+1) = C_j \frac{v_{Cj,N}(k+1) - v_{Cj,N}(k)}{T_s}\Big|_{j=+,-}, \tag{13}$$

$$\begin{aligned} v_{C+,N}(k+1) \cdot S_{1n}(k+1) &+ v_{C-,N}(k+1) \cdot (1 - S_{1n}(k+1)) \\ &= R_{L_N} \cdot i_{L_N}(k+1) + L_N \frac{\Delta i_{L_N}(k+1)}{T_s} \end{aligned}, \tag{14}$$

$$\begin{aligned} i_{C-}(k+1) + i_{C+}&(k+1) + i_{L_N}(k+1) = \\ &- \sum_{j=a,b,c} i_{out,j}(k+1) - \sum_{j=a,b,c} i_{C_f,j}(k+1) \cdot \end{aligned} \tag{15}$$

The values of the inductor currents of the LC-filter ($i_{L_{fj}}(k+1)$) can be predicted with the following equations:

$$\begin{aligned} v_{out,jN}(k+1) &= v_{C+}(k+1) \cdot S_{1j}(k+1) \\ &+ v_{C-}(k+1) \cdot (1 - S_{1j}(k+1)) - L_f \frac{\Delta i_{L_{fj}}(k+1)}{T_s}, \\ &- R_{L_f} \cdot i_{L_{fj}}(k+1)\big|_{j=a,b,c} \end{aligned} \tag{16}$$

$$i_{C_{fj}}(k+1) = C_f \frac{\Delta v_{out,j}(k+1) - R_{C_f}\Delta i_{C_{fj}}(k+1)}{T_s}, \tag{17}$$

$$i_{grid,j}(k+1) + i_{\mu grid,j}(k+1) = i_{L_f,j}(k+1) - i_{C_f,j}(k+1). \tag{18}$$

Under the assumption $Z_n^{th}(k+1) = Z_n^{th}(k)$ and by approaching the equivalent Thevenin's impedance only for the fundamental frequency, the relationship (19) can be obtained in case that $X_j^{th}(k) \geq 0$ and (20) when $X_j^{th}(k) < 0$:

$$\begin{aligned} v_{PCC,j}(k+1) &= R_j^{th,\mu grid}(k) \cdot i_{\mu grid,j}(k+1) \\ &+ L_j^{th,\mu grid}(k) \frac{i_{\mu grid,j}(k+1) - i_{\mu grid,j}(k)}{T_s}\Bigg|_{j=a,b,c} \end{aligned}, \tag{19}$$

$$\begin{aligned} i_{\mu grid,j}(k+1)\Big|_{j=a,b,c} &= \\ C_j^{th,\mu grid}(k) &\frac{\Delta[v_{PCC,j}(k+1) - R_j^{th,\mu grid}(k) \cdot i_{\mu grid,j}(k+1)]}{T_s} \cdot \end{aligned} \tag{20}$$

In those cases, when the inverter works tied to the main grid:

$$\begin{aligned} v_{grid,j}(k+1) - v_{PCC,j}(k+1) &= R^{grid} \cdot i_{grid,j}(k+1) \\ &+ L_j^{grid} \frac{i_{grid,j}(k+1) - i_{grid,j}(k)}{T_s}\Bigg|_{j=a,b,c} \end{aligned}. \tag{21}$$

### 2.3. Cost Function for the Islanded Mode

In this working mode, the inverter object of this study has to manage the voltage waveform in which concerns to magnitude, frequency, harmonics content and phase equilibrium. In order to achieve these criteria, the cost function expressed in (22) is divided into three main parts: $J_{isl}^{wave}$ which manages the waveform of the output voltage, $J_{isl}^{harm}$ which minimizes the harmonics content and $J_{isl}^{bal}$ which controls the balance between phases. In order to use the predictive model of the inverter, the assumption that between two sample instants $Z_{out,\alpha}^{th}(k+1) = Z_{out,\alpha}^{th}(k)$ has to be used in the predictive model of the inverter.

$$\min_{\mathbf{s}(k)} J_{isl}(k) = \min_{\mathbf{s}(k)} \left( J_{isl}^{wave}(k) + J_{isl}^{harm}(k) + J_{isl}^{bal}(k) \right), \tag{22}$$

$$
\begin{aligned}
J_{isl}^{wave}(k) = \sum_{\alpha=a,b,c} \Bigg[ & w_{isl}^{inst} \left( v_{out,\alpha}(k+1) - v_{out,\alpha}^{ref}(k+1) \right)^2 \\
& + w_{isl,\alpha}^{cycle} \left( \Re e(\mathscr{U}_{out,\alpha}(k+1)) - \Re e(\mathscr{U}_{out,\alpha}^{ref}(k+1)) \right)^2 \\
& + w_{isl,\alpha}^{cycle} \left( \Im m(\mathscr{U}_{out,\alpha}(k+1)) - \Im m(\mathscr{U}_{out,\alpha}^{ref}(k+1)) \right)^2 \Bigg]
\end{aligned} \tag{23}
$$

At each sample instant, the voltage reference is calculated and imposed in the first term of (23), minimizing the difference between the predicted voltage and the calculated reference. In order to minimize the steady-state error the second term of (23) is added, correcting this error with the complete fundamental cycle computation.

$$
\begin{aligned}
J_{isl}^{harm}(k) = \sum_{\alpha=a,b,c} \Bigg[ & w_{isl,\alpha}^{v}(\Delta v_{out,\alpha}(k+1))^2 \\
& + w_{isl,\alpha}^{i}(\Delta i_{out,\alpha}(k+1))^2 \Bigg] \\
& + w_{isl}^{cap}(v_{C+}(k+1) - v_{C-}(k+1))^2
\end{aligned} \tag{24}
$$

The first and second term of (24) minimize the voltage and current abrupt variations between two sample instants, avoiding the harmonics content in both voltage and current. The third term manages the balance of voltage for the neutral point.

$$J_{isl}^{bal}(k) = \sum_{\alpha=a,b,c}^{\beta=b,c,a} w_{isl}^{bal}(|\mathscr{U}_{out,\alpha}(k+1))| - |\mathscr{U}_{out,\beta}(k+1)|)^2. \tag{25}$$

When unbalanced loads are connected to the inverter the voltage magnitude between phases is uncompensated. In order to control these situations, the term expressed in (25) is included in the cost function.

### 2.4. Cost Function for the Grid-Connected Mode

In the grid connected mode, it is assumed that due to the fact that the voltage reference is imposed by the main grid. Under the assumption of robustness in the voltage waveform provided by the main grid and considering that Park's transformation is a rotational reference frame, it is considered that between two sample instants the dqo-voltage is constant. Under this assumption and using the predictive model, the output currents $i_{out,\gamma}(k+1)$ $i_{grid,\gamma}(k+1)$ of each phase can be obtained.

The controller receives the set-point for the exchange of active and reactive powers with the main grid ($P_{grid,\alpha}^{ref}(k+1)$, $Q_{grid,\alpha}^{ref}(k+1)$). Due to the fact that $\mathscr{U}_{grid,\alpha}^{ref}(k)$ is imposed by the main grid and supposed constant between two sample instant, the reference current $\mathscr{I}_{grid,\alpha}^{ref}(k)$ can be easily obtained with the following equations:

$$P_{grid,\alpha}^{ref}(k) = \frac{|\mathscr{U}_{grid,\alpha}^{ref}(k)||\mathscr{I}_{grid,\alpha}^{ref}(k)|}{2} \cos(\varphi_{grid,\alpha}^{ref}(k)), \tag{26}$$

$$Q_{grid,\alpha}^{ref}(k) = \frac{|\mathscr{U}_{grid,\alpha}^{ref}(k)||\mathscr{I}_{grid,\alpha}^{ref}(k)|}{2} \sin(\varphi_{grid,\alpha}^{ref}(k)). \tag{27}$$

The current references are calculated as follows:

$$i_{grid,\alpha}^{ref}(k+1) = \\ |\mathscr{I}_{grid,\alpha}^{ref}(k+1)| \sin(\omega(k+1+D_\alpha) + \varphi_{grid,\alpha}^{ref}(k+1)) \tag{28}$$

A digital delay $D_\alpha$ has to be included, which is adaptive with $Z_{out,\alpha}^{th}(k)$. As done for the case of islanded mode, the cost function in the grid-connected mode is divided into three parts:

$$\min_{\mathbf{s}(k)} J_{conn}(k) = \min_{\mathbf{s}(k)} \Big( J_{conn}^{wave}(k) + J_{conn}^{harm}(k) + J_{conn}^{bal}(k) \Big), \tag{29}$$

$$J_{conn}^{wave}(k) = \sum_{\alpha=a,b,c} \Big[ w_{conn}^{inst} \Big( i_{grid,\alpha}(k+1) - i_{grid,\alpha}^{ref}(k+1) \Big)^2 \\ + w_{conn,\alpha}^{cycle} \Big( \Re e(\mathscr{I}_{grid,\alpha}(k+1)) - \Re e(\mathscr{I}_{grid,\alpha}^{ref}(k+1)) \Big)^2 \\ + w_{conn,\alpha}^{cycle} \Big( \Im m(\mathscr{I}_{grid,\alpha}(k+1)) - \Im m(\mathscr{I}_{grid,\alpha}^{ref}(k+1)) \Big)^2 \Big] \tag{30}$$

The procedure to formulate (30) is similar to the one carried out for (23). At each sample instant, the current reference is calculated and imposed in the first term of (30), minimizing the difference between the predicted current exchanged with the main grid and the reference calculated. In order to minimize the steady state error, the second term of (30) is added, correcting this error with the complete fundamental cycle calculation done for the current exchange with the main grid expressed in Fourier's domain.

$$J_{conn}^{harm}(k) = \sum_{\alpha=a,b,c} \Big[ w_{conn,\alpha}^{v} (\Delta v_{out,\alpha}(k+1))^2 \\ + w_{conn,\alpha}^{i} \Big( \Delta i_{grid,\alpha}(k+1) \Big)^2 \Big] \\ + w_{isl}^{cap} (v_{C+}(k+1) - v_{C-}(k+1))^2 \tag{31}$$

The second part of the cost function in grid-connected mode (31) minimizes the harmonic injection in the current to the grid, as well as the voltage variations in the microgrid. It also balances the neutral point of the inverter. Finally, when unbalanced loads are connected to the microgrid, they can affect to balance in the active and reactive power injected to the main grid. For this purpose, the term of the cost function expressed in (32) is included.

$$J_{conn}^{bal}(k) = \\ \sum_{\alpha=a,b,c}^{\beta=b,c,a} w_{conn}^{bal}(P_{grid,\alpha}(k+1)) - P_{grid,\beta}(k+1)))^2 \\ + \sum_{\alpha=a,b,c}^{\beta=b,c,a} w_{conn}^{bal}(Q_{grid,\alpha}(k+1)) - Q_{grid,\beta}(k+1)))^2 \tag{32}$$

### 2.5. Cost Function for the Interconnected Mode

The interconnected mode can be considered as a hybrid mode between the connected and the islanded mode, since a main grid which imposes the references in voltage and frequency does not exist, but there can be energy exchange between the interconnected

microgrids. Due to the fact that there is not a main grid, both microgrids have to work controlling the voltage and the frequency, the so-called multi-master mode.

$$
\min_{\mathbf{s}(k)} J_{inter}^{(X)}(k) = \min_{\mathbf{s}(k)} \Big( J_{isl}^{(X),wave}(k) + J_{isl}^{(X),harm}(k)
$$
$$
+ J_{isl}^{(X),bal}(k) + \sum_{\gamma=a,b,c} (i_\gamma^{(X)->(Y)}(k))^2 \Big). \tag{33}
$$

The notation $(X)$ refers to the microgrids (A) and (B) and the terminology $(X)->(Y)$ makes reference to the exchange between the microgrid (X) and the microgrid (Y), being $i_{(X)->(Y)}^{exch}$ the exchanged current between the microgrid (X) and the microgrid (Y). Notice that this term achieves to synchronize in frequency both microgrids and also to equilibrate the voltage magnitude between both microgrids without being necessary any kind of communication between the interconnected microgrids.

## 3. Simulation Results

The simulations are carried out using MATLAB/Simukink/Simpower©. The simulation is run each $T_{simulation} = 1$ μs. The controller acts with a sample time of $T_s = 20$ μs. The different values for the simulation and power inverter components are exposed in Table 1.

**Table 1.** Components value.

| Parameter | Value |
| --- | --- |
| Filter inductance $L_f$ | 1 [mH] |
| Filter inductance resistance $R_{L_f}$ | 0.1 [Ω] |
| Filter capacitor $C_f$ | 0.5 [mF] |
| Filter capacitor resistance $R_{C_f}$ | 0.1 [Ω] |
| DC link voltage $U_{dc}$ | 950 [V] |
| Neutral inductance $L_N$ | 2.5 [μF] |
| Neutral inductance resistance $R_{L_N}$ | 0.1 [Ω] |
| Neutral balancing capacitors $C_+, C_-$ | 6600 [μF] |
| Grid connection line inductance $L_{grid}$ | 0.1 [mH] |
| Grid connection line resistance $R_{grid}$ | 0.1 [Ω] |
| Slave inverter line inductance $L_{inv}$ | 0.1 [mH] |
| Slave inverter line resistance $R_{inv}$ | 0.1 [Ω] |
| Non-linear load line inductance $L_{non}$ | 0.1 [mH] |
| Non-linear load line resistance $R_{L_{non}}$ | 0.1 [Ω] |
| Non-linear load dc resistance $R_{non}$ | 60 [Ω] |
| Non-linear load dc capacitor $C_{non}$ | 6.6 [mF] |
| Unbalanced load phase a resistance $R_a$ | 1 [MΩ] |
| Unbalanced load phase b resistance $R_b$ | 10 [Ω] |
| Unbalanced load phase c resistance $R_c$ | 10 [Ω] |
| Unbalanced load phase b inductance $L_b$ | 1 [mH] |
| Unbalanced load phase c capacitor $C_c$ | 0.1 [mF] |

The different weighting factors exposed in the different cost functions are obtained using an adaptive strategy. The main concept pursued has been to create a virtual LC filter able to minimize the harmonic content in current or voltage (depending of the working mode) according to the equivalent Thevenin's impedance seen by the controller. With this aim, the controller was tested simulating different impedances (injecting or consuming current from the microgrid) with different values of module $|Z_{th}|$ and angle $|\theta_{th}|$ using the inverter connected to the PCC through $L_{inv}$ and $R_{inv}$. For each impedance, different values of the weighting factors were tested selecting those ones that achieved best transient response and lower harmonics content. With this procedure a heuristic law is obtained for each weighting factor as function of the equivalent module and angle of $Z_{th}$ seen by the

inverter object of study. Later on, the simulation provided in the following sections were used in order to verified the result of these heuristic laws obtained for the weighting factors.

### 3.1. Comparison between MPC and PI-PWM Controllers for Single Microgrids

The first simulation is used to compare the results in both grid-connected and islanded modes, as well as the transition between modes using an MPC-controller and a PI-PWM-controller for a single microgrid working in both modes: grid-connected and islanded. In this simulation the non-linear and the unbalanced loads are connected to the microgrid in all the sample instants. Both controllers receive the next references for the power exchange with the main grid: $[P^{ref}_{grid,\alpha}, Q^{ref}_{grid,\alpha}] = [-15{,}000 \text{ W}, -9000 \text{ Var}] \; \forall t \in [0 \text{ s}, 0.5 \text{ s}]$ and $[P^{ref}_{grid,\alpha}, Q^{ref}_{grid,\alpha}] = [+15{,}000 \text{W}, +9000 \text{ Var}]$ at $\forall t \geq 0.5$ s. Between $t \in [1 \text{ s}, 1.5 \text{ s}]$ a fault in the main grid occurs so the transition to islanded mode is required, restoring the connection of the microgrid with the main grid for $t > 1.5$ s. The comparison between the results obtained in the reference tracking for the active and reactive power between the MPC and the PI-PWM controller can be found in Figure 3. As can been seen in the figure, the PI-PWM controller presents a longer transient response while the MPC controller reaches the given references in just two cycles of the fundamental frequency. In Figure 4, the comparison between the THD results for the MPC and PI-PWM controller are exposed. As can be seen, despite the presence of non-linear and unbalanced loads, the current waveforms present a low harmonic content in the MPC-controller while the PI-PWM controller is not able to minimize the current harmonic content. During the instants $t = 1$ s and $t = 1.5$ s, a grid blackout occurs and the power inverter works in islanded mode. The comparison between the behavior of the power inverter with the MPC and the PI-PWM controllers can be seen in Figures 5 and 6, where the voltage magnitude and the phase values are shown. As it occurs for the case of grid-connected mode, a better transient response is obtained in the case of the MPC controller. A better response is also obtained for THD values of the voltage at the PCC in the case of the MPC controller, as shown in Figure 7.

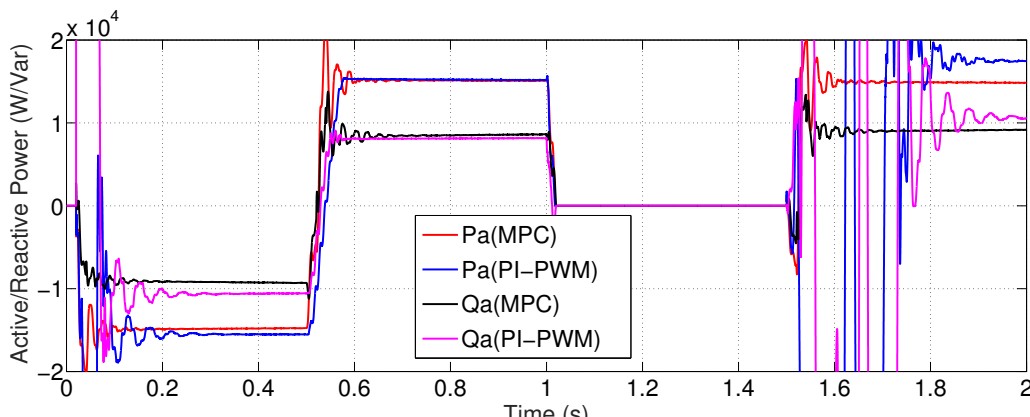

**Figure 3.** Comparison of the results for the active and reactive power exchange with the main grid between the Model Predictive Control (MPC) and PI-PWM controllers for phase *a*.

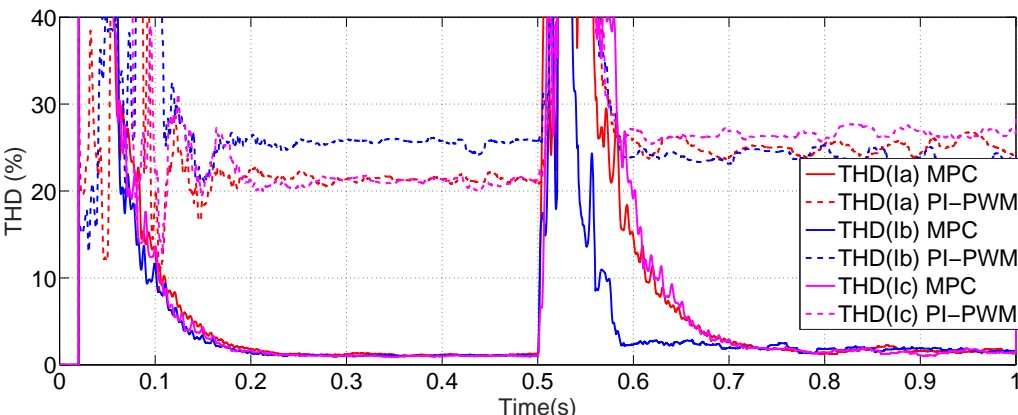

**Figure 4.** Comparison of the THD values for the current exchange with the main grid between the MPC and PI-PWM Controllers.

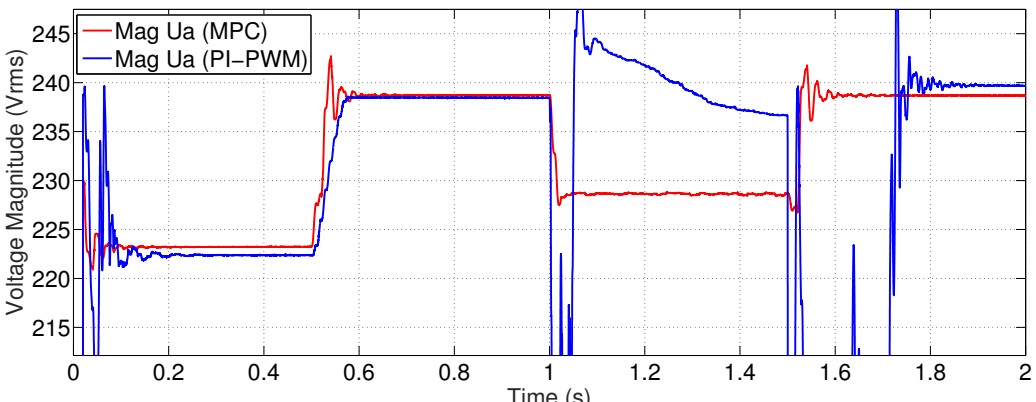

**Figure 5.** Voltage Magnitude for phase A at the PCC.

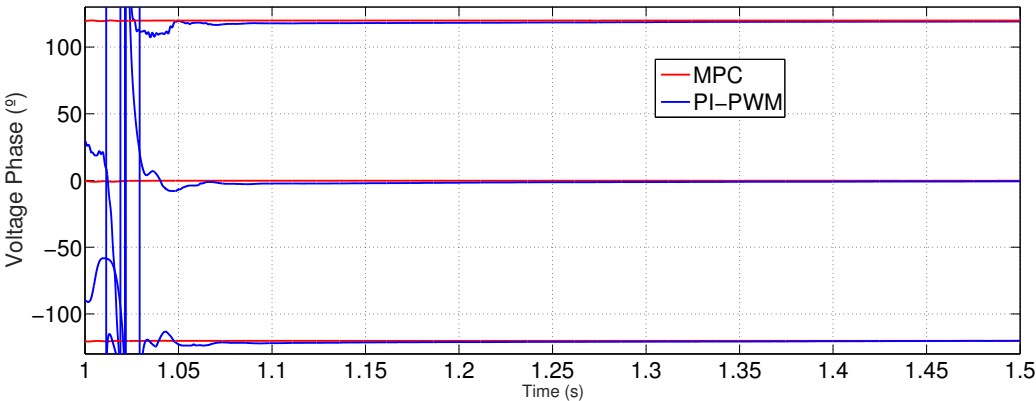

**Figure 6.** Absolute voltage phase angle value of the voltages at the Point of Common Coupling (PCC) during the blackout of the main grid.

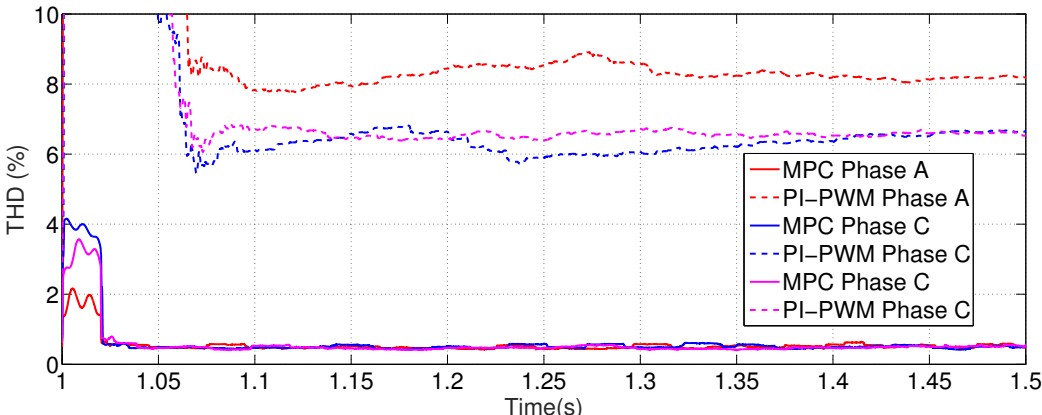

**Figure 7.** THD values for the voltages at the PCC during the blackout of the main grid.

*3.2. Power Quality Management Results for Interconnected Microgrids Working without Presence of Grid*

The aim of the second simulation launched is to evaluate the behavior of the presented controller for the case of interconnected microgrids working under a grid blackout. In this case, the IPS-A and IPS-B are connected and IPS-grid is disconnected (see Figure 1). In the case of the microgrid (A) the non-linear loads are connected during all the sample instants of the simulation and the unbalanced loads are connected for these sample instants $t > 0.2$ s. In the microgrid (B) the unbalanced loads are connected during all the sample instants and the non-linear loads are connected at $t > 0.1$ s. In Figures 8 and 9, a comparison between the obtained results for the voltage magnitudes for every phase of each microgrid is shown. The corresponding phase and THD values are displayed in Figures 10 and 11. The current consumption can be observed in Figure 12. As can be seen, for the sample instants $t \in [0.10, 0.12]$ in Figure 9, a more robust behavior is obtained when working interconnected, where the voltage magnitudes of each microgrid are always $|\mathcal{U}_{out,\gamma}^{(X)}| > 200$ for both microgrids. In Figure 13, the obtained results for the current exchange between both microgrids are shown. As can be seen, each microgrid manages its own loads without nearly non-affection to the neighbor microgrid. As can be seen in Figures 10 and 11, the presence of non-linear and unbalanced loads and the changes in current demand at each microgrid, as well as the interaction between microgrids, do not affect the THD content in voltage nor the balance between phases guaranteeing the power quality supply to the loads connected to both microgrids.

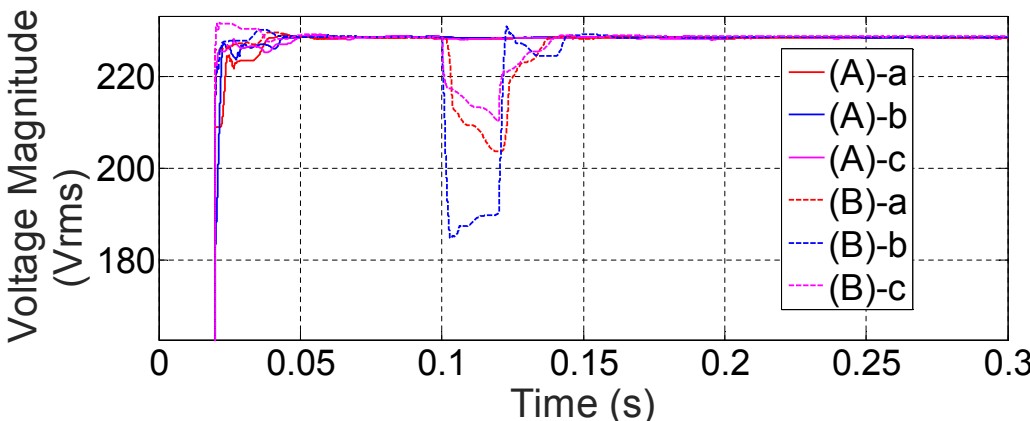

**Figure 8.** Voltage Magnitude per phase and microgrid in mode non-interconnected and grid-islanded.

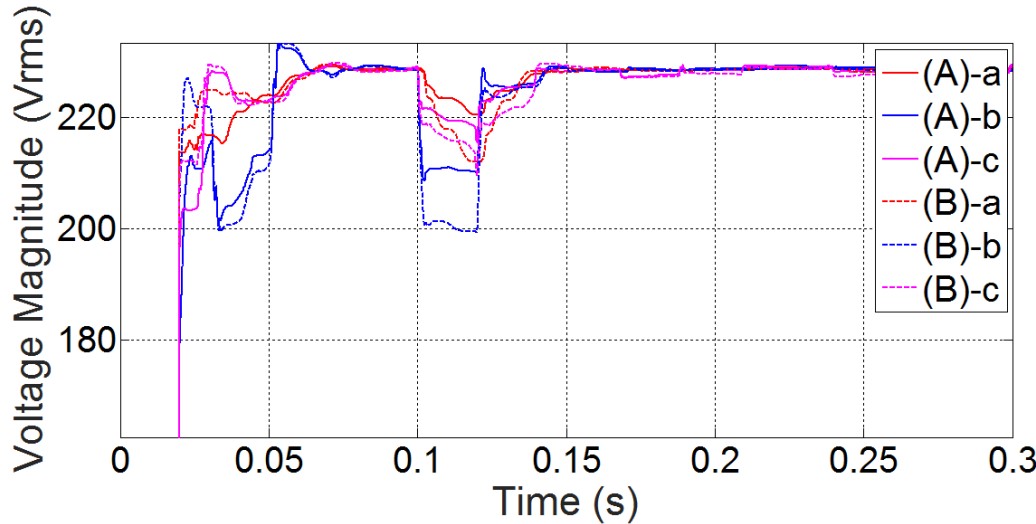

**Figure 9.** Voltage Magnitude per phase and microgrid in mode interconnected and grid-islanded.

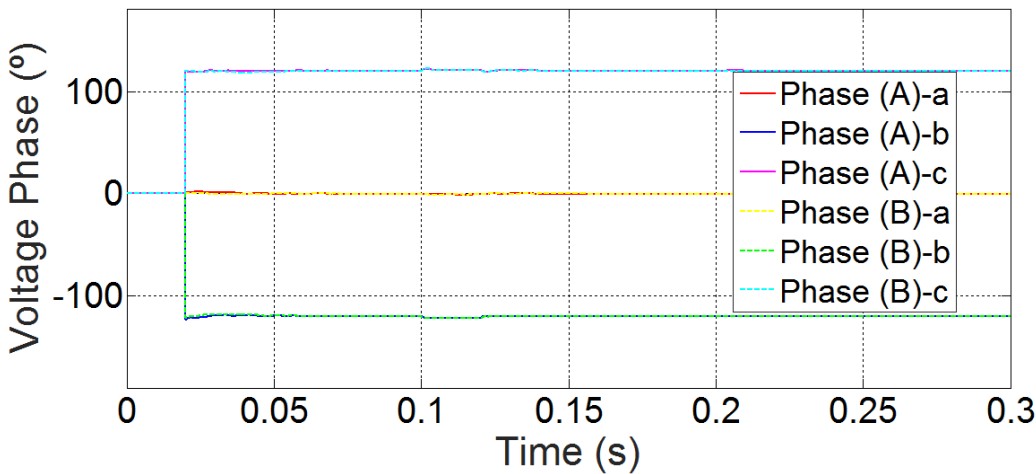

**Figure 10.** Absolute voltage phase angle value per phase and microgrid in mode interconnected and grid-islanded.

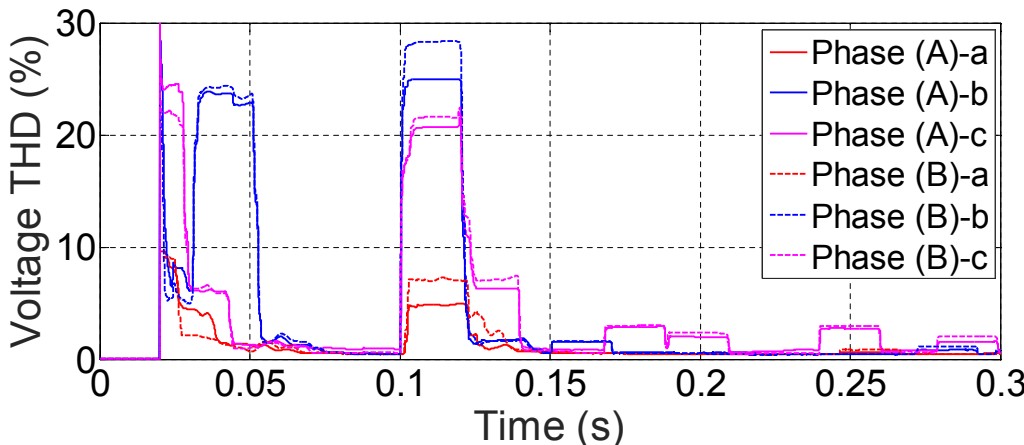

**Figure 11.** Voltage THD per phase and microgrid in mode interconnected and grid-islanded.

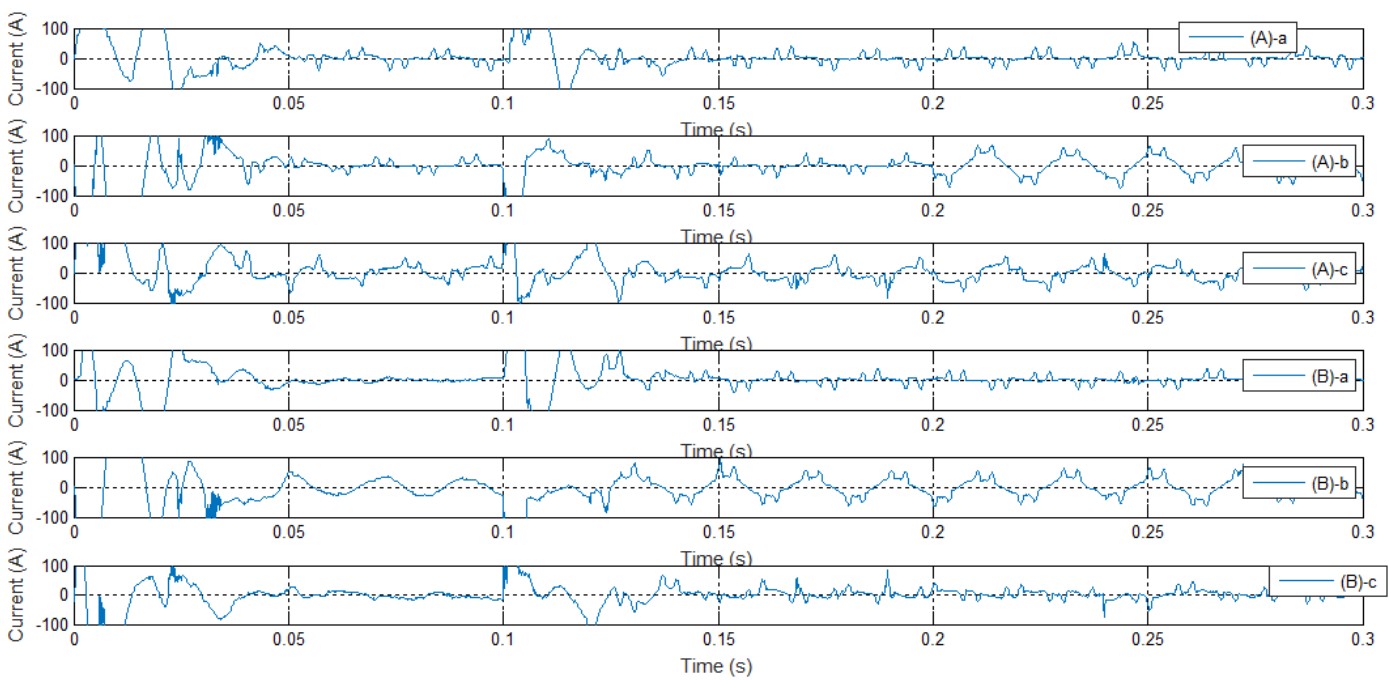

**Figure 12.** Current per phase and microgrid in mode interconnected and grid-islanded.

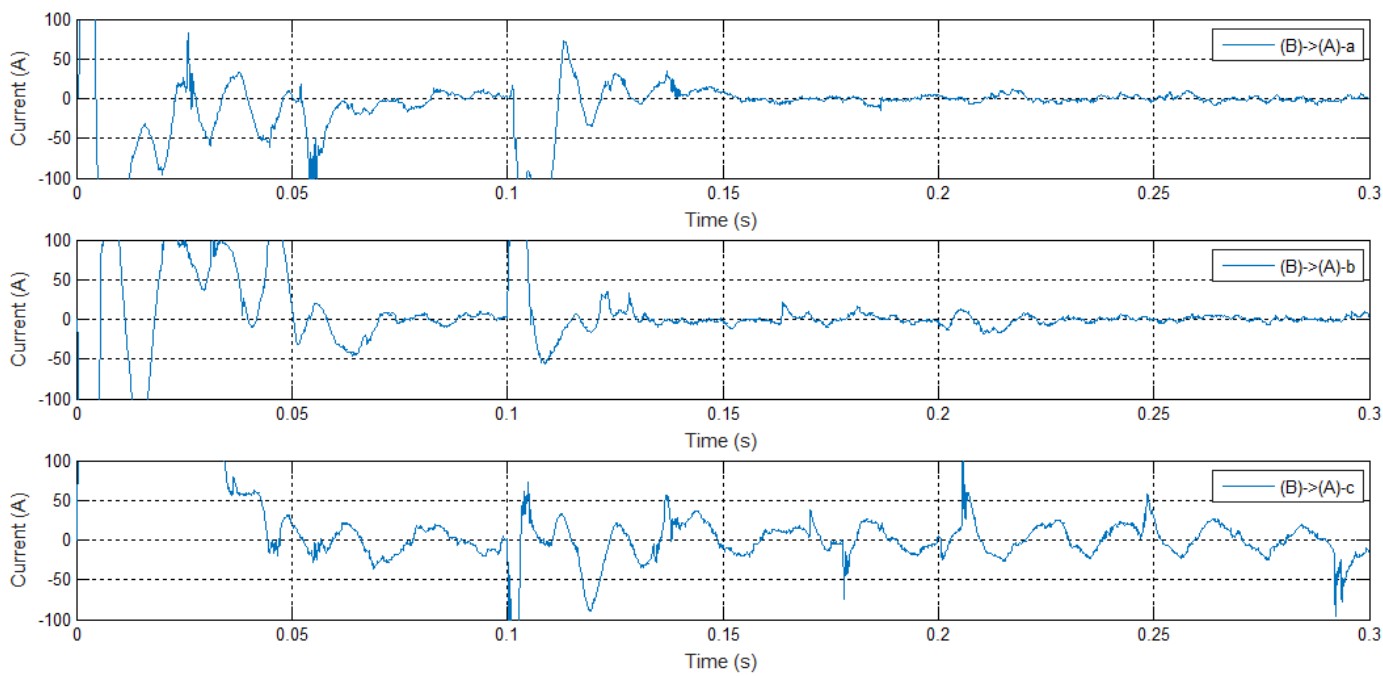

**Figure 13.** Current exchange per phase between microgrid (A) and microgrid (B).

## 4. Conclusions

In the present study, the behavior of a new algorithm applied to control a four-wire three-phase VSI with active control of the neutral point which governs a microgrid in both modes grid-connected and islanded has been exposed. It has been developed using the finite-state MPC control technique with a control horizon equal to "1". The results show an optimal behavior for the output variables of the inverter, with a low THD in voltage in the case of islanded and in the current exchanged with the grid when working as a grid-tied inverter. The difference with previous works is the use of the mean voltage/power values evaluated in the fundamental component. This allows controlling even with the harmonic

presence correcting the low horizon prediction limitation that MPC applied to power electronic has. With this method, although the inverter needs to be modeled, the load is modeled on-line by the controller with accurate results for the current prediction in all the exposed cases with the developed technique. The inverter has been designed to act as a master of a microgrid. The most critical cases as non-linear sources, non-linear loads and unbalanced loads have been tested showing an accurate response for each one of the exposed cases. A fast transition behavior when it is required to switch the working mode is also found. The results shown that beside the non-linearities and unbalances found in the microgrid, the inverter accomplishes with the standard EN-50160 for the islanded mode regrading voltage harmonics content. Despite the presence of unbalanced and non-linear loads, it also fulfills what corresponds with the standard IEC 61000-3-2 and IEC 61000-3-4 regarding the harmonic current emission limits for balanced system.

The controller has been also validated for the implementation to manage the power quality in interconnected microgrids acting when they are grid-connected or under a grid blackout where they have to work interconnected but islanded from the main grid. The control algorithm is based on a MPC-controller applied to a four-wire three-phase VSI with active control of the neutral point which works as master of a microgrid with unbalanced and non-linear loads and generators connected. The simulation results show the potential of the presented MPC-controller in comparison with classical PI-PWM controllers solving the transient response problems of traditional methods. The fact of possessing an accurate transient response is specially advantageous for power quality problems in microgrids overall if unbalanced and non-linear loads and generators are connected to the microgrids. As can be seen, the developed methodology is improved with its application to the case of interconnected microgrids acting islanded from the main grid.

**Author Contributions:** Conceptualization, All; methodology, All; software, F.G.-T. and S.V.; validation, F.G.-T. and S.V.; formal analysis, All; investigation, All; resources, All; data curation, All; writing—original draft preparation, All; writing—review and editing, All; visualization, All; supervision, All; project administration, All; funding acquisition, All. All authors have read and agreed to the published version of the manuscript.

**Funding:** This research has been funded by European Commission with the European Regional Development Funds (ERDF) under the program Interreg SUDOESOE3/P3/E0901 (Project IMPROVEMENT-Integration of combined cooling, heating and power microgrids in zero-energy public buildings under high power quality and continuity of service requirements).

**Conflicts of Interest:** The authors declare no conflict of interest. The funders had no role in the design of the study; in the collection, analyses, or interpretation of data; in the writing of the manuscript, or in the decision to publish the results.

## Abbreviations

The following abbreviations are used in this manuscript:

| | |
|---|---|
| CCS | Continuous Control Set |
| DER | Distributed Energy Resources |
| DMPC | Distributed Model Predictive Control |
| ESS | Energy Storage System |
| FCS | Finite Control State |
| GPC | Generalized Predictive Control |
| MPC | Model Predictive Control |
| PQR | Power Quality and Realibility |
| RES | Renewable Energy System |
| SHE | Selective Harmonic Elimination |
| SP | Smith Predictor |
| VSI | Voltage Source Inverter |

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
