# Peer review of "Microgrids Power Quality Enhancement Using Model Predictive Control"

_electronics, doi:10.3390/electronics10030328_

Round 1

Reviewer 1 Report

Throughout the document, please edit for English language clarity.

line 1: Briefly define “power quality” for context.

line 3: What are the requirements for this system?

line 6: Indicate here that this is a comparison study utilizing simulations.

line 15: What are the different national policies?

line 25: What is the high resistive component?

line 82: Please clarify what is meant by “sampling instant” in the context of this sentence.

Section 1.1: The paragraphs in this section are very long and should be split into multiple paragraphs.

line 191-192: Define the “islanded and grid-connected operation modes” here in this paragraph.

Figure 1: To provide context, all resistor, capacitor, inductor, transistor, and all other designators should be listed in a table along with an appropriate description so that an overview is provided to help the reader understand how the model is constituted.  This could be an updated version of Table 1. In the caption of Figure 1, indicate whether the passive components are assumed to be ideal.

Page 6: “As is well-known” should be removed from the sentence; this is superfluous.

line 226: Indicate how J_isl “manages the waveform of the output voltage”

Page 9: What is the “sample instant”?  Indicate the sampling frequency here of the simulation or associated (conceptual) ADC and frontend system that would be used in an actual prototype.  Perhaps indicate that this is given in Section 3?

Line 236: What is meant by “who imposes the references”?  Please clarify.

line 290: Clarify what is meant by with a control horizon equal than "1".  Is the control horizon equal or not equal to one?

Author Response

The authors are very grateful for the kind review process carried out by Reviewer 1. The authors have tried to integrate all the comments providing a new version of the paper. The different changes carried out have been highlighted using the color blue.

Throughout the document, please edit for English language clarity.

line 1: Briefly define “power quality” for context.

line 3: What are the requirements for this system?

line 6: Indicate here that this is a comparison study utilizing simulations.

The abstract has been modified according to these three comments.

line 15: What are the different national policies?

Section 1 has been reduced according to the third comment of reviewer 3. This sentence has been removed

line 25: What is the high resistive component?

Section 1 has been reduced according to the third comment of reviewer 3. This sentence has been removed. The authors referred to a resistive behavior of this kind of loads.

line 82: Please clarify what is meant by “sampling instant” in the context of this sentence.

This sentence has been replaced by the following one:

<<Every time that the controller is executed, it enumerates the set of admissible switching sequences, predicting the corresponding system response based on a prediction model and evaluating the cost function according to the prediction carried out. The controller applies to the system the control sequence which yields the minimal value in the cost function>>

Section 1.1: The paragraphs in this section are very long and should be split into multiple paragraphs.

Section 1.1 has been modified according to this comment. Just to justify that it has been also reduced to answer the third reviewer’s comment.

line 191-192: Define the “islanded and grid-connected operation modes” here in this paragraph.

 We have included the main functionalities of the islanded and grid-connected mode in the new version of the abstract. As reviewer 3 has specified to reduce this section, we consider the most appropriate is not to repeat the definition of both modes in Section 1.1. Nevertheless, if reviewer 1 will require this definition we will provide it in the following version of the paper.

Figure 1: To provide context, all resistors, capacitors, inductors, transistors, and all other designators should be listed in a table along with an appropriate description so that an overview is provided to help the reader understand how the model is constituted.  This could be an updated version of Table 1. In the caption of Figure 1, indicate whether the passive components are assumed to be ideal.

The next clarifying sentence has been added in the new version of the paper in Section 2 – Controller Design and highlighted in color blue:

<<…The power inverter is composed of four-legs $(a,b,c,n)$, composed each one by two ideal power switches, one connected to the positive terminal of the dc voltage source and the other switch connected to the negative terminal. The switches states are a function of the associated gate-signals. The gate signals applied to the switches situated in the same leg of the inverter are related, being the value of the negative terminal connected gate signal the opposed to the positive terminal gate signal. Each gate signal just can adopt two values: ”0” if the power switch is at OFF-state and ”1” when the power switch is at ON-state. To integrate the possibility to manage unbalanced and non-linear loads, the neutral point of the power inverter is connected to two similar capacitors $C_+$ and $C_-$. The voltage of $C_+$ and $C_-$ is controlled with the power switches $S_{1,n}$ connected to the positive terminal of the $V_{dc}$ source and a common point with $S_{2,n}$ which is connected to the negative terminal. The common point between  $S_{1,n}$ and $S_{2,n}$ is connected with the neutral point of the 2L-VSI through a non-ideal inductance composed by $R_{L_N}$ and $L_N$. 

The power inverter is connected in the Point of Common Coupling (PCC) of the microgrid with the power grid and the rest of the components of the microgrid through an LC filter, being $C_f$ is the capacitor of the filter, $L_f $is the inductance of the filter. To integrate the non-ideal behavior of these components a small $R_{L_f}$ is the series resistance of the inductance while $R_{C_f}$ is the associated resistance of the capacitor $C_f$. The power inverter feeds a microgrid composed of an unbalanced load, a non-linear load, and a bidirectional inverter. The unbalanced load is composed by a resistor $R_a$ in phase $a$, a non-ideal inductance formed by $R_b$ and $L_b$ in phase $b$, and finally, it has got connected to a non-ideal capacitor in phase c whose components are $R_c$ and $C_c$. The non-linear load is formed by an uncontrolled rectifier and a load on the DC side formed by a capacitor filter $C_{non}$ and the resistor $R_{non}$. The rectifier is connected with the PCC of the microgrid with the line impedances given by $L_{non}$. Finally, an  AC/DC bidirectional 2L-VSI with LC filter is also included connected to microgrid through the line impedance given by $R_{inv}$ and $L_{inv}$. >>

Page 6: “As is well-known” should be removed from the sentence; this is superfluous.

It has been removed.

line 226: Indicate how J_isl “manages the waveform of the output voltage”
Mainly with its first term given in expression (23). As explained in Section 1.1, << …FCS is based on the finite number of switching states that a power inverter can adopt. The optimization problem is simplified with the prediction of the converter behavior considering these possible switching states.  Every time that the controller is executed, it enumerates the set of admissible switching sequences, predicting the corresponding system response based on a prediction model and evaluating the cost function according to the prediction carried out. The controller applies to the system the control sequence which yields the minimal value in the cost function>>

So according to the different output predictions based on the predictive model of the inverter and equivalent Thevenin’s impedance calculated by the microgrid, the controller executes the gate-signal combination which minimizes (22).

Page 9: What is the “sample instant”?  Indicate the sampling frequency here of the simulation or associated (conceptual) ADC and frontend system that would be used in an actual prototype.  Perhaps indicate that this is given in Section 3?

 It is indicated in Section nevertheless, the given sentence was not enough clarifying so the following one is exposed in the new version of the paper

<<…The simulations are carried out using Simpower$^{\copyright}$. The simulation is run each  $T_{simulation} = 1\mu s$. The controller acts with a sample time of $T_{s} = 20\mu s$>>

Line 236: What is meant by “who imposes the references”?  Please clarify.

It is detailed in the following sentence:

<<… Since there is not a main grid both microgrids have to work controlling the voltage and the frequency, the so-called multi-master mode.>>

line 290: Clarify what is meant by a control horizon equal to "1".  Is the control horizon equal or not equal to one?

Thanks for this comment, as well as the rest. Our apologies for all these mistakes. The control horizon is equal to one. It has been also modified in the text.

Reviewer 2 Report

The paper presents a model-based control scheme for inverters employed in microgrids, designed to provide improved power quality during transients following disturbances, such as blackout and reconnection. The main novelty is in the use of Fourier transforms to evaluate instantaneous active and reactive power in real time. Detailed model of the microgrid is employed in a model predictive control scheme, through minimisation of defined cost functions to determine an optimal switching state at each sampling instant.

The presented results are encouraging, but the authors should address the following questions.

  1. Performance of the Fourier transform in this context is somewhat unclear, particularly in view that it is calculated in real time and under transient conditions. Presumably FFT is employed, so how big is the sampling record, what windowing function is used, and how does this relate to the transient nature of the cases analysed? What is the measurement time delay introduced by the chosen sample length? The authors only present standard equations for the continuous Fourier transform and provide no mention of FFT nor considerations for its use in this scenario.

  1. The paper makes no assessment of the sensitivity of the proposed method with respect to model accuracy, so robustness of the proposed scheme is questionable. The authors need to provide an adequate analysis of these aspects. Overall, the control method appears to rely on cancellation of plant dynamics on the basis of the model, without a clear mechanism for accommodating modelling errors, so it is likely to suffer from robustness issues. It is not clear what mechanism in the proposed scheme ensures that the states of the plant model do not diverge from those of the physical plant.

  1. Implementation details of the proposed control scheme are not provided. What precisely is the algorithm used? What optimisation method is employed for cost function minimisation? What is the computational complexity and is it feasible for real-time implementation?

The manuscript should be rechecked to improve English expression and correct typos.

Author Response

The paper presents a model-based control scheme for inverters employed in microgrids, designed to provide improved power quality during transients following disturbances, such as blackout and reconnection. The main novelty is in the use of Fourier transforms to evaluate instantaneous active and reactive power in real-time. A detailed model of the microgrid is employed in a model predictive control scheme, through minimization of defined cost functions to determine an optimal switching state at each sampling instant.

The presented results are encouraging, but the authors should address the following questions.

First of all, the authors would like to thank the consideration of the paper carried by Reviewer 2, as well as his kind comments which have allowed us to improve our work. We try to answer all his comments, highlighting the changes in the paper using a blue color.  

  1. The performance of the Fourier transform in this context is somewhat unclear, particularly in view that it is calculated in real-time and under transient conditions. Presumably, FFT is employed, so how big is the sampling record, what windowing function is used,

The FFT method was not employed. It has been used the discrete expression of the Fourier’s transform. The sampling records correspond to a full fundamental cycle of 0.02 s (50 Hz) discretized in periods of Ts=20us, 1000 positions.

and how does this relate to the transient nature of the cases analyzed?

The transient nature is given by the difference between the sampling instant used for the simulation Ts=1 us and the one used in the controller Ts=20us.

 What is the measurement time delay introduced by the chosen sample length?

Just a time delay of a sample instant is introduced. The controller assumes that the equivalent Thevenin’s impedance see at the output filter of the inverter is similar to that at the previous sample instant. This delay is similar i.e PI-PWM controllers.

The authors only present standard equations for the continuous Fourier transform and provide no mention of FFT nor considerations for its use in this scenario.

To clarify, the next sentence has been included and highlighted in the new version paper:

<<…using the  discrete expressions of equations

 (1)-(3) with a sample period $T_s$, for the voltage and current signals,

 the value of this signal expressed in the Fourier's domain…>>

  1. The paper makes no assessment of the sensitivity of the proposed method with respect to model accuracy, so the robustness of the proposed scheme is questionable. The authors need to provide an adequate analysis of these aspects. Overall, the control method appears to rely on the cancellation of plant dynamics on the basis of the model, without a clear mechanism for accommodating modelling errors, so it is likely to suffer from robustness issues. It is not clear what mechanism in the proposed scheme ensures that the states of the plant model do not diverge from those of the physical plant.

The authors agree with this comment. The authors have not available the physical model of the inverter, so just can provide the simulation results. For this reason, a perfect model is considered. In other experiences with physical inverters, this is carried out using a cascade controller structure with a PI-controller with modifies the voltage and current references given to the MPC controller, to take into account the contained error of the model.

  1. Implementation details of the proposed control scheme are not provided. What precisely is the algorithm used? What optimisation method is employed for cost function minimisation?

The method is based on the Finite Control Set MPC. We highlight this issue with the following paragraphs along with the paper:

Section 1.1. Literature Review

<<The FCS-MPC methodology is based on the finite number of switching states that a power77inverter can adopt.   The optimization problem is simplified with the prediction of the converter78behavior considering these possible switching states. Every time that the controller runs, the set of79admissible switching sequences are numbered, thus predicting the corresponding system response80based on a prediction model and evaluating the cost function according to the prediction carried81out. The controller applies to the system the control sequence, which yields the minimal value in the82cost function.  Therefore the cost function is minimized using the Exhaustive Searching Algorithm83method [11].>>

Section 2. Controller Design

<< The first step of the controller is to calculate the Fourier analysis of the current and voltage output at the current sample instant $k$. With these measurements, the Thevenin's equivalent impedance is calculated at the output of the inverter. With this equivalent impedance, the output current and voltage prediction are carried out which is included in the cost function to be minimized.  Finally, the optimal gate-signal combination is calculated in the FCS-MPC controller>>

Therefore the methodology to minimize the cost function is just to select the best gate signal combination.

What is the computational complexity and is it feasible for real-time implementation?

The computational complexity is not high, since the algorithm just consists of the equations provided and the evaluation of 16 different possibilities of combination of gates. Similar algorithms have been applied for MPC controllers using just an OPAL-RT platform 4500 or a DSP of Texas Instruments 28335.

The next sentence is included in Section II. Controller Design.

<<Note that the current control hardware platform provides a high computational capability which makes the proposed method feasible as can be a observe in [26].>>

The manuscript should be rechecked to improve English expression and correct typos.

The authors apologize and agree with this comment. An exhaustive English revision has been carried out. Nevertheless, if the English would not be good enough authors will agree to pay a native revision.

Reviewer 3 Report

Comments to the authors

Manuscript number: electronics-1079365

Title: Microgrids Power Quality Enhancement Using Model Predictive Control

1) Avoid using unclear terms. In abstract, the authors claim that the developed method “shows an optimal behavior”. What do you mean by the optimality? With respect to what?

2) One issue of in interconnected microgrids (e.g., hybrid AC/DC microgrids) is the power loss and power quality degradation due to the circulating current. This issue has been discussed in 10.1049/iet-rpg.2014.0271 and 10.1109/TIE.2006.882019. The authors need to discuss this very important issue in their paper.

3) The reviewer appreciate the provided literature review. However, Section 1 occupies one-third of the manuscript (5 pages out of 17). It might be a good idea to revise Section 1.

4) I fail to understand why the authors have considered a AC-DC converter connected to the PCC. Is it used to connected DC loads?

5) There is a confusion in the time stamps. What is the difference between t and tk? What is k? Does k represent prediction?

6) No discussion on the value of weights (w) in (23)-(25) is resented. How can one determine these weights? The same for weights in (30)-(32).

7) The value of all parameters and the configuration of the software should be provided for reproduction purposes.

8) According to Figure 8, though the developed MPC scheme can reduce the THD, it causes a high overshoot during the transient.

Author Response

First of all, the authors apologize for the detected error introduced in the first draft of the manuscript. The authors completely agree with all the comments addressed by reviewer 3 being very grateful for his/her help to improve our paper.

1) Avoid using unclear terms. In the abstract, the authors claim that the developed method “shows optimal behavior”. What do you mean by optimality? With respect to what?

 A new version of the abstract has been provided. These terms have been removed addressing the abstract to the comparison with classical PI methods.

2) One issue of interconnected microgrids (e.g., hybrid AC/DC microgrids) is the power loss and power quality degradation due to the circulating current. This issue has been discussed in 10.1049/iet-rpg.2014.0271 and 10.1109/TIE.2006.882019. The authors need to discuss this very important issue in their paper.

It is modeled with the grid impedance of each microgrid Lgrid and Rgrid, see Figure 1.

Authors included this important discussion with the following sentence in the literature review including both references:

<<…The importance of the management of power losses and power quality degradation due to the circulating current in interconnected microgrids (e.g., hybrid AC/DC microgrids) is studied in [20] and [21].>>

3) The reviewer appreciate the provided literature review. However, Section 1 occupies one-third of the manuscript (5 pages out of 17). It might be a good idea to revise Section 1.

According to this comment Section 1 has been reduced.

4) I fail to understand why the authors have considered an AC-DC converter connected to the PCC. Is it used to connected DC loads?

The authors agree it has not been enough clarified. A wide description of Figure 1, has been included according to a comment requested by Reviewer 1. The next clarifying sentence has been added in the new version of the paper in Section 2 – Controller Design and highlighted in color blue:

<<…The power inverter is composed of four-legs $(a,b,c,n)$, composed each one by two ideal power switches, one connected to the positive terminal of the dc voltage source and the other switch connected to the negative terminal. The switches states are a function of the associated gate-signals. The gate signals applied to the switches situated in the same leg of the inverter are related, being the value of the negative terminal connected gate signal the opposed to the positive terminal gate signal. Each gate signal just can adopt two values: ”0” if the power switch is at OFF-state and ”1” when the power switch is at ON-state. To integrate the possibility to manage unbalanced and non-linear loads, the neutral point of the power inverter is connected to two similar capacitors $C_+$ and $C_-$. The voltage of $C_+$ and $C_-$ is controlled with the power switches $S_{1,n}$ connected to the positive terminal of the $V_{dc}$ source and a common point with $S_{2,n}$ which is connected to the negative terminal. The common point between  $S_{1,n}$ and $S_{2,n}$ is connected with the neutral point of the 2L-VSI through a non-ideal inductance composed by $R_{L_N}$ and $L_N$. 

The power inverter is connected in the Point of Common Coupling (PCC) of the microgrid with the power grid and the rest of the components of the microgrid through an LC filter, being $C_f$ is the capacitor of the filter, $L_f $is the inductance of the filter. To integrate the non-ideal behavior of these components a small $R_{L_f}$ is the series resistance of the inductance while $R_{C_f}$ is the associated resistance of the capacitor $C_f$. The power inverter feeds a microgrid composed of an unbalanced load, a non-linear load, and a bidirectional inverter. The unbalanced load is composed by a resistor $R_a$ in phase $a$, a non-ideal inductance formed by $R_b$ and $L_b$ in phase $b$, and finally, it has got connected to a non-ideal capacitor in phase c whose components are $R_c$ and $C_c$. The non-linear load is formed by an uncontrolled rectifier and a load on the DC side formed by a capacitor filter $C_{non}$ and the resistor $R_{non}$. The rectifier is connected with the PCC of the microgrid with the line impedances given by $L_{non}$. Finally, an AC/DC bidirectional 2L-VSI with LC filter is also included connected to microgrid through the line impedance given by $R_{inv}$ and $L_{inv}$. >>

 The rectifier is used to introduce a non-harmonic free load. The bidirectional inverter has been used for weighting-parameters-adjust purposes. Please, see Reviewer 3’s sixth comment.

5) There is a confusion in the timestamps. What is the difference between t and tk? What is k? Does k represent prediction?

The authors apologize for these nomenclature errors. In the new version, t is used for the continuous-time, k for the discrete instants, and k+n for the predictions.  

6) No discussion on the value of weights (w) in (23)-(25) is resented. How can one determine these weights? The same for weights in (30)-(32).

The authors also agree with this comment. To clarify this the next sentence has been included in the new version of the paper.

<<… The different weighting factors exposed to the different cost functions are obtained using an adaptive strategy. The main concept pursued has been to create a virtual LC filter ables to minimize the harmonic content in current or voltage (depending on the working mode) according to the equivalent Thevenin’s impedance seen by the controller. With this aim, the controller was tested simulating different impedances (injecting or consuming current from the microgrid) with different values of module $|Z_{th}|$ and angle $|\theta_{th}|$ using the inverter connected to the PCC through $L_{inv}$ and $R_{inv}$. For each impedance, different values of the weighting factors were tested selecting those that achieved the best transient response and lower harmonics content. With this procedure, a heuristic law is obtained for each weighting factor as a function of the equivalent module and angle of $Z_{th}$ seen by the inverter object of study. Later on, the simulation provided in the following sections was used to verify the result of these heuristic laws obtained for the weighting factors.>>

7) The value of all parameters and the configuration of the software should be provided for reproduction purposes.

All the parameters have been included in Table I and the configuration of the software in the paragraph included at the beginning of Section 3.

8) According to Figure 8, though the developed MPC scheme can reduce the THD, it causes a high overshoot during the transient.

From the author’s knowledge, currently, this overshoot is produced with all the controllers during the transition between grid-connected and islanded mode. The proposed methodology is just 0.02 s. As can be seen in the rest of the figures this overshoot is worst if traditional PI-PWM controllers are used.

Round 2

Reviewer 3 Report

No further comment.